# Sweet Potatoes Puree Mixed with Herbal Aqueous Extracts: A Novel Ready-to-Eat Product for Lactating Mothers

**Luiza-Andreea Tănase (Butnariu)** [1], **Doina-Georgeta Andronoiu** [1], **Oana-Viorela Nistor** [1,*], **Gabriel-Dănuț Mocanu** [1], **Elisabeta Botez** [1] and **Bogdan Ioan Ștefănescu** [2]

[1] Faculty of Food Science and Engineering, "Dunărea de Jos" University of Galați, 111 Domnească Street, 800201 Galați, Romania; luiza.tanase@ugal.ro (L.-A.T.)
[2] Faculty of Medicine and Pharmacy, "Dunărea de Jos" University of Galați, 35 Al. I. Cuza Street, 800010 Galați, Romania
\* Correspondence: oana.nistor@ugal.ro

**Abstract:** Worldwide, around 385 thousand babies are born each day. Many of them cannot be breastfed because of several physiological problems of the mothers. Galactogogues remain the most natural and prolific way to improve both milk quantity and quality. Various herbs are traditionally used to increase lactation, but the best known are fennel (*Foeniculum vulgare* L.) and anise (*Pimpinella anisum* L.). The main objective of the present study was to obtain some special and nutritious ready-to-eat products from pureed sweet potato (*Ipomoea batatas* L.) fortified with aqueous extracts from the aforementioned galactogogues herbs. Two different types of thermal treatment, steaming and baking, were investigated to obtain healthy and safe-for-consumption purees. Steam convection had a lower impact, compared with hot air convection, on the content of bioactive compounds among all samples. Among all samples, sweet potato puree with fennel aqueous extract, processed by steaming, (EFCA) showed the highest content of β-carotene (1.27 ± 0.11 mg/g DW), lycopene (0.59 ± 0.07 mg/g DW), and total carotenoids (1.38 ± 0.11 mg/g DW); the cooking loss registered statistically significantly lower values in the case of steam convection. These reports might potentially generate novel ready-to-eat foods used as meals and as well as lactation adjuvants.

**Keywords:** herbs; sweet potatoes; FT-IR; phytochemical content; DSC

## 1. Introduction

Sweet potatoes, or *Ipomoea batatas*, belong to the *Convolvulaceae* family. South America is the region in which it was initially grown for its tubers. *I. batatas* are currently commonly grown in a multitude of tropical, subtropical, and warm environments. The tuber of *I. batatas* is abundant in starch, dietary fiber, pectin, minerals, vitamins, and bioactive compounds. The main bioactive phenolic compounds are caffeoylquinic acid derivatives (CQA derivatives), such as chlorogenic acid (5-CQA), 3,4-; 3,5- and 4,5-di-caffeoylquinic acid (-diCQA); 3,4,5-tri-caffeoylquinic acid (3,4,5-triCQA); and feruloyl-caffeoylquinic acid (FCQA). The roots of *I. batatas* and the aerial parts of the tuber are both used as nourishment. Moreover, it has therapeutic elements that are utilized in traditional medicine to treat a wide variety of diseases (cancer, diabetes, and cardiovascular disease) [1,2]. It is a productive and adaptable crop but also a primary source of starch. Due to its special nutritional and functional properties, sweet potatoes are widely cultivated and industrialized in order to sustain human health [3]. Due to its high polyphenolic content, sweet potatoes are a particularly excellent source of antioxidants, especially orange varieties whom provide a significant source of phenolic acids [4]. The cumulative effect of phytochemicals from nutritious vegetables and beneficial herbs could sustain the production of special destination ready-to-eat products. According to Sibeko et al. [5], a significant proportion of the global population continues to be confident of herbal medicines for their primary healthcare needs, with women traditionally using herbs for lactation and in the postpartum period. Actual

postpartum practices vary between cultures, even though the majority of directly linked scientific papers reflect only a possible preventive approach. The few species identified in scientifical publications, from industrialized nations, lack perspective on earlier patterns of use or their health influence. In a survey conducted by Barnes et al. [6], it is noted that the perceived improvements to the health of the expectant or breastfeeding child were the primary reasons for pregnant and breastfeeding women to use lactation adjuvants. The majority of the respondents had previously used them and believed that their actions had induced significant health improvements. Although, respondents preferred to take medications prescribed by their doctors or pharmacists over complementary/herbal medicine products.

Foods are physically complex products, with rheological and material properties that are time and process dependent. Thermal treatments were chosen to ensure safe consumption and preservation, but also for the development of taste and flavor [7]. Comparing hot air convection to water vapor convection, Richardson [8] stated that airflow is much more prone to the development of turbulence than liquid flows, due to the fact that higher Reynolds numbers are achieved. Therefore, turbulence increases mixing and surface heat transfer rates in the case of baking. On the other hand, steam convection uses lower temperatures (60–100 °C) and consumes less energy.

This study aims to improve the use of galactogogue herbs, such as fennel (*Foeniculum vulgare* L.) and anise (*Pimpinella anisum* L.), in different ready-to-eat purees, especially designed for pregnant or breastfeeding women. The objectives of this work were to obtain various ready-to-eat products mixed with herbal aqueous extracts using two different thermal treatments, followed by the study of their influence over numerous important scientific aspects such as bioactive compounds, color parameters, texture, sensory attributes, and others.

The novelty of the study consists of using such a valuable plant-based material, namely sweet potato, known for its nutritional benefits, especially vitamins and fibers.

## 2. Materials and Methods

### 2.1. Reagents and Chemicals

2,2-diphenyl-1-picryhydrazyl (DPPH), 6-hydroxy-2,5,7,8-tetramethylchromane-2-carboxylic acid (Trolox), potassium persulfate ($K_2O_2S_8$), Folin–Ciocâlteu reagent, gallic acid, sodium carbonate ($Na_2CO_3$) 20%, quercetin, sodium nitrite ($NaNO_2$) 5%, aluminum chloride ($AlCl_3$) 10%, sodium hydroxide (NaOH) 1M, methanol (HPLC grade), hexane (HPLC grade), and acetone (HPLC grade) were used for the samples analysis, as mentioned before by Tanase (Butnariu) et al. [9].

### 2.2. Sample Preparation

#### 2.2.1. Aqueous Extracts of Galactogogue Herbs

Aqueous extracts of the used herbs were obtained using the method described by Butnariu et al. [10]. Briefly, 5 g of each herb was grinded and mixed with 125 mL of bidistilled water and boiled on water bath for 30 min, then filtered, cooled, and refrigerated (4 °C).

#### 2.2.2. Preparation of Sweet Potato Puree Mixed with Herbal Aqueous Extract

Sweet potatoes (4–5 cm diameter) were collected from a supermarket (Galați, Romania) on the day the experiments were conducted. They were peeled, washed, and chopped into rings 2 cm in heigh, which were cut into 4. Subsequently, two types of heat treatment were used for processing, being hot air convection (baking) at 180 °C for 35 min (electric oven Indesit FIMB-51K.A-IX-PL, Poland) and steaming (steam cooker Zelmer 37Z010, Poland) at 94 °C for 15 min. The processing time was established and dependent on the temperature of at least 80 °C, specific the pasteurization.

The sweet potatoes experimentally reached these temperatures; for the hot air convection treatment, the core temperature of the sweet potatoes was 93.6 ± 1.62 °C whereas for

steaming was 89.5 ± 0.26 °C, respectively. To turn them into puree, the processed potatoes were firstly pureed by using a vertical mixer (Bosch ErgoMixx, Göcklingen, Germany) for 2 min at 1900 rpm before being mixed with herbal aqueous extract of anise or fennel (6%) and salt (0.5%).

Samples encoding is presented as follows:

$CM_1$ and $CM_2$ are purees treated by steam and hot air convection, respectively.

EFCA and EACA are steamed purees mixed with fennel or anise aqueous extract.

EFCC and EACC are baked purees mixed with fennel or anise aqueous extract.

### 2.3. Heat Induced Changes

2.3.1. Cooking Loss Measurements and Calculation

The samples' mass was measured before and after applying the heat treatment. Cooking loss and cooking yield were calculated according to Sengun et al. [11], using the following equations:

$$\text{Cooking loss}(\%) = \frac{\text{Uncooked sample weight} - \text{Cooked sample weight}}{\text{Uncooked sample weight}} \times 100 \quad (1)$$

$$\text{Cooking yield}(\%) = \frac{(\text{Cooked sample weight})}{(\text{Uncooked sample weight})} \times 100 \quad (2)$$

2.3.2. Differential Scanning Calorimetry (DSC)

The DSC method used in order to determine thermic measurements was detailed by Tanase (Butnariu) et al. [9] in a previous study. Each sample was placed into 160 µL aluminum crucibles with pins and lids. One cooling–heating cycle from −20 °C to 150 °C was applied. Five segments with a cooling rate of 10 K/min correlated with the heating rate were used as follows: 6 K/min between −20 °C/50 °C, 2 K/min between 50 °C/80 °C, 1 K/between 80 °C/150 °C, and two isothermal processes of 2 min duration at each −20 °C, and 150 °C temperature, respectively.

### 2.4. Phytochemical Characterization

For the determination of antioxidant activity by DPPH assay [12], total phenolic content (TPC) [13] and total flavonoid content (TFC) [14], methanol 70% extraction was performed, while *n*-hexane: acetone (ratio 3:1 *v/v*) was used for β-carotene, lycopene, and total carotenoids quantification. Briefly, for all the determinations, an amount of 1 g of sample was dissolved in 10 mL of a mixture of methanol 70% or *n*-hexane: acetone. After the extraction, the samples were centrifuged at 9000× *g*, 4 °C for 5 min before further analyses (antioxidant activity, TPC, and TFC) or before being spectrophotometrically read. The contents of carotenoids were spectrophotometrically read at the specific wavelengths for total carotenoids, β-carotene, and lycopene, and calculated according to Escoto et al. [15].

### 2.5. Determination of the In Vitro Release of Phenols from Sweet Potato Purees

In vitro digestion was performed as described by Minekus et al. [16] simulating both gastric and intestinal phases. The entire digestion process consisted of incubating the samples at 37 °C under shaking at 150 rpm. An orbital shaking incubator (Medline Scientific, Chalgrove, Oxon, UK) was used and the sampling was made at every 30 min.

### 2.6. Color Evaluation of Sweet Potato Purees

The main color parameters (L*, a* and b*) were measured using a Minolta Chroma Meter CR-410 (Konica Minolta, Osaka, Japan). The calculation methods for the total color difference (ΔE), browning (BI), yellowness (YI), and whiteness (WI) were based on the equations presented by [9]. All the tests were performed in triplicate for fresh and processed samples at $T_0$ and $T_7$, respectively.

### 2.7. Rheological Analysis

A control-stress rheometer AR2000ex (TA Instruments, New Castle, DE, USA) was used to perform the rheological measurements, which allows temperature control due to the integrated heating and cooling Peltier plate. Small amplitude tests, namely, strain sweep, and frequency sweep tests, were applied between two parallel plates (20 mm diameter) with a gap of 1 mm [17]. During the strain sweep test, a strain between 0.01 and 100% was applied at 1 Hz frequency. For the frequency sweep test, an oscillation between 0.1 and 100 Hz was applied. In addition, three interval thixotropy test (3ITT) was used to identify the thixotropic behavior of samples [18]. To calculate the recovery degree of the samples, Equation (3) was used.

$$\%Rec = \frac{G'_{10}}{G'_1} \times 100 \tag{3}$$

In Equation (3), $G_1'$ is the value of elastic modulus at the beginning of the determination and $G_{10}'$ is the value of elastic modulus after 10 min. Two determinations for each sample were made.

### 2.8. Instrumental Determination of Texture

A Brookfield CT3 Texture Analyzer was used to investigate the texture of the sweet potatoes puree. The packaging used were plastic vessels with a diameter of 35 mm and a height of 50 mm. The samples were subjected to a double penetration test with an acrylic cylinder (24 mm diameter). Penetration speed was set at 1 mm/s and target distance was set at 10 mm [19]. Textural parameters were determined using the TexturePro CT V1.5 software. Three measurements were made for each sample.

### 2.9. FT-IR Spectroscopy

The infrared spectra were collected using a Nicolet iS50 FT-IR spectrometer (Thermo Scientific, Cincinnati, OH, USA) equipped with a built-in ATR accessory, DTGS detector, and KBr beamsplitter. Therefore, 32 scans were co-added over the range of 4000–400 cm$^{-1}$ with a resolution of 4 cm$^{-1}$, and the method was fully described previously by Tanase (Butnariu) et al. [9].

### 2.10. Sensory Analysis

Ten untrained panelists from the authors affiliation faculty, aged from 25 to 55 years old and all non-smokers, performed sensory analysis at room temperature (21 ± 2 °C). Before taking part in the sensorial analysis, participants were given a thorough explanation of the sample manufacturing and processing method and its consumption advantages. Each panelist received the samples in transparent plastic containers which had been coded randomly. A hedonic nine-point scale, with 1 being the weakest/unpleasant perception and 9 being the strongest/most pleasant perception, was used to mark the intensity of each attribute. The tasters identified the following sensory attributes: overall and exterior acceptance, firmness, consistency, cohesiveness, color, taste, aroma, aftertaste, and mouthcoating.

The ethic approval was obtained from the Ethical Committee of "Dunărea de Jos" University of Galati, Romania, No. 141/CEU from 27 March 2023.

### 2.11. Statistical Analysis

One-way analysis of variance (ANOVA) and Tukey test were used to identify significant differences ($p < 0.05$) between the samples (Minitab 17 statistical software). Data are provided as mean ± standard deviation (SD), after each analysis was performed at least twice.

## 3. Results and Discussion

### 3.1. Heat-Induced Changes

3.1.1. Cooking Loss Measurements and Calculation

Cooking loss is a genuine phenomenon of thermal processing, which can occur from the loss of some main compounds denaturized by the time–temperature correlation associated with water evaporation. In a study of Pan et al. [20], it is mentioned that steaming process might have a negative effect on retaining the total content of carotenoids in sweet potatoes, which can have a direct involvement in cooking loss and cooking yield.

Besides the main purpose, the cooking yield was evaluated in order to choose the most feasible method to produce purees with consistent quality, despite the several variations due to cultivar differences, variation in bioactive compounds content, type of processing, and storage.

Similar results for the cooking loss were obtained for steam convection by Phan et al. [21] in a study of sweet potatoes originating from Vietnam, while for baking the result was higher with almost 25% of this value presented in Table 1, possibly induced by the protection given by the natural peel.

**Table 1.** Influence of hot air and steam convection on raw sweet potato.

| Parameters | Baking | Steaming |
|---|---|---|
| Cooking loss, % | $36.88 \pm 0.08$ [A] | $5.72 \pm 0.31$ [B] |
| Cooking yield, % | $63.12 \pm 0.08$ [B] | $94.28 \pm 0.32$ [A] |

The averages on the same line with different superscripts (A–B) are statistically significantly different ($p < 0.05$).

Subsequently, moisture content was retained due to the natural barrier created by the superficial layer of cells dehydrated during baking.

The results from Table 1 present a higher cooking yield ($94.28 \pm 0.32\%$) for steam convection compared with hot air convection, which is expected due to the properties of this type of processing. The cooking yield for hot air convection ($63.12 \pm 0.08\%$) is dependent on the free water content of sweet potatoes, which was evaporated during the thermal processing and on the processing time exposure (35 min).

3.1.2. Differential Scanning Calorimetry (DSC)

The thermal properties of a product are important functional properties involved in many processing phenomena; thus, it is of high importance to know the sweet potato matrix behavior during processing.

The lower initial freezing temperature of $-20\,^{\circ}\text{C}$ is a result of increased solid content, mostly sugars, which act as plasticizers to increase the amount of water which is unavailable for freezing and reduce the amount of energy required to freeze the processed sweet potato [22].

Figure 1 presents the variation of heat flow as a function of temperature from $-20$ to $+150\,^{\circ}\text{C}$. The heat flow has registered negative values between $-182.07$ mW and $-2.32$ mW.

It could be observed that the melting area of the sweet potato is present between 80.83 and $112.33\,^{\circ}\text{C}$. An endotherm peak is slightly marked at $104.16\,^{\circ}\text{C}$ for sweet potato roots. The corresponding energy needed to produce the melting transition of the organics from the sweet potato is $-53.50$ mW. In a study of Ahmed et al. [23], several temperature values for the interaction of starch granules and water are identified, similar to these from the present study, which are responsible for the amorphous background region.

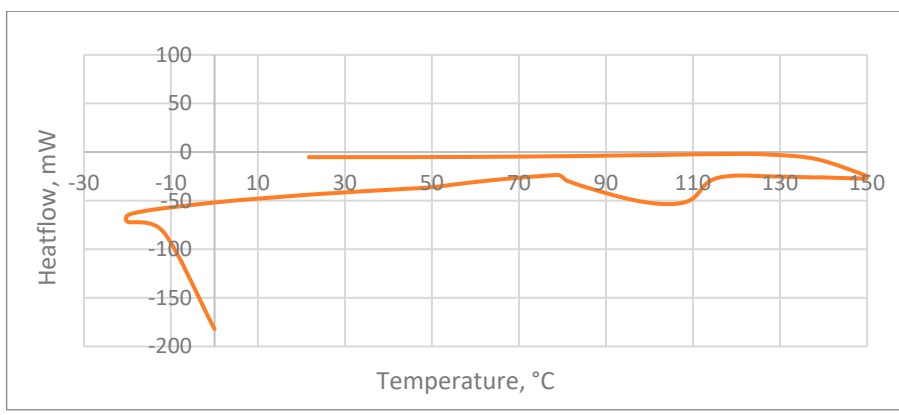

**Figure 1.** DSC thermograms of raw sweet potato samples.

*3.2. Phytochemical Content*

The total phytochemical content of several sweet potato purees, improved with the herbal aqueous extracts, is presented in Table 2.

**Table 2.** Phytochemicals content and antioxidant activity of sweet potatoes samples mixed with herbal aqueous extracts.

| Phytochemicals and Antioxidant Activity of Sweet Potato Samples | $T_0$—Samples in the First Day of Storage | | | | | |
|---|---|---|---|---|---|---|
| | Steaming | | | Baking | | |
| | CM$_1$ | EFCA | EACA | CM$_2$ | EFCC | EACC |
| Antioxidant Activity, µM Trolox/g DW | 35.4 ± 0.70 A,B | 40.72 ± 2.24 A | 40.52 ± 0.48 A | 30.96 ± 1.64 B | 36.95 ± 0.85 A | 35.85 ± 3.75 A,B |
| TPC, mg GAE/g DW | 4.59 ± 0.18 B | 5.03 ± 0.09 A | 5.04 ± 0.19 A | 3.38 ± 0.10 D | 3.73 ± 0.04 C,D | 3.85 ± 0.17 C |
| TFC, mg EQ/g DW | 5.98 ± 0.46 A,B,C | 6.56 ± 0.17 A | 6.28 ± 6.28 A,B | 4.73 ± 0.09 D | 5.29 ± 0.18 C,D | 5.63 ± 0.19 B,C |
| BC, mg/g DW | 0.68 ± 0.09 B,C | 1.27 ± 0.11 A | 0.81 ± 0.09 B | 0.47 ± 0.03 D | 0.56 ± 0.01 C,D | 0.6 ± 0.02 C,D |
| LYC, mg/g DW | 0.45 ± 0.02 B | 0.59 ± 0.07 A | 0.5 ± 0.05 A,B | 0.28 ± 0.03 C | 0.33 ± 0.01 C | 0.32 ± 0.02 C |
| TC, mg/g DW | 0.72 ± 0.04 B | 1.38 ± 0.11 A | 0.82 ± 0.10 B | 0.46 ± 0.03 C | 0.5 ± 0.02 C | 0.64 ± 0.01 B,C |
| | $T_7$—Samples after 7 Days Refrigeration Storage | | | | | |
| | Steaming | | | Baking | | |
| Antioxidant Activity, µM Trolox/g DW | 28.9 ± 1.35 D | 40.69 ± 0.43 B | 33.68 ± 0.93 C | 32.44 ± 0.55 C | 44.97 ± 1.23 A | 46.5 ± 1.29 A |
| TPC, mg GAE/g DW | 6.68 ± 0.05 A | 6.71 ± 0.11 A | 6.8 ± 0.04 A | 3.55 ± 0.01 C | 5.32 ± 0.18 B | 5.45 ± 0.09 B |
| TFC, mg EQ/g DW | 6.1 ± 0.25 B | 6.8 ± 0.04 A | 7.15 ± 0.18 A | 3.97 ± 0.03 D | 5.52 ± 0.04 C | 5.39 ± 0.07 C |
| BC, mg/g DW | 0.67 ± 0.01 D | 1.14 ± 0.04 A | 1.01 ± 0.04 B | 0.43 ± 0.01 E | 0.78 ± 0.02 C | 0.67 ± 0.00 D |
| LYC, mg/g DW | 0.48 ± 0.01 C | 0.61 ± 0.03 A | 0.54 ± 0.03 B | 0.23 ± 0.01 E | 0.46 ± 0.01 C | 0.4 ± 0.00 D |
| TC, mg/g DW | 0.97 ± 0.01 C | 1.18 ± 0.02 A | 1.06 ± 0.04 B | 0.43 ± 0.01 F | 0.78 ± 0.02 D | 0.69 ± 0.00 E |

The averages on the same line with different superscripts (A–F) are statistically significantly different ($p < 0.05$).

As shown in Table 2, steam convection surpassed hot air convection in preserving the phytochemical content of sweet potato purees. Therefore, the sample with fennel aqueous extract, EFCA, processed by steam convection, registered the highest values for the overall phytochemical characterization ($p < 0.05$).

According to Xu et al. [2], polar compounds (such as phenolic acids) might be partially degraded during a longer time of heating, as in the case of hot air convection compared to steaming.

β-carotene is an orange-yellowish pigment found in plants and an antioxidant that protects the cells from the production of free radicals [24]. The β-carotene content registered by all the samples, both treated by water vapor and hot air convection, was at least three

times higher compared to a study conducted by Oloniyo et al. [25] on three different types of sweet potatoes, processed into flour after boiling and drying, where the content of β-carotene varied between 0.03 to 0.18 mg/g DW. On the other hand, a study conducted by Zhang et al. [26] on microwaved assisted thermally sterilized sweet potato puree registered similar values as our control steamed sample (CM$_1$). Therefore, the increase of β-carotene values in the enhanced samples is attributed to the herbal aqueous extract addition.

The content of lycopene and total carotenoids was statistically higher ($p < 0.05$) in all the steamed samples, as mentioned earlier, especially in those enriched with fennel or anise aqueous extract. The degradation of these compounds during hot air convection might be generated due to the chemical reactions such as β-carotene isomerization and lycopene oxidation [27,28]. Similarly, in a study regarding the boiling, steaming, and baking of sweet potatoes [29], it was shown that baking significantly reduced the total carotenoid and β-carotene content values compared to the other two procedures.

For all the determined phytochemical contents, the samples recorded higher or comparable values with the fresh samples after one week of storage at refrigeration temperature (4 °C). Therefore, the shelf life of such ready-to-eat products may benefit from this. In consequence, there were samples whose β-carotene content increased and others that showed a slight decrease. The reduction of β-carotene content might be due to the high residual oxygen content from each polyacrylic cylinder containing a sample during storage, which was different for each sample due to handling. According to Zhang et al. [26], as the remaining oxygen inside the packaging decrease, there is no further degradation of β-carotene.

### 3.3. Determination of the In Vitro Release of Remanent Phenolic Content from Sweet Potatoes Purees

Figure 2 represents the gastrointestinal behavior of polyphenols throughout a 4 h simulation of digestion.

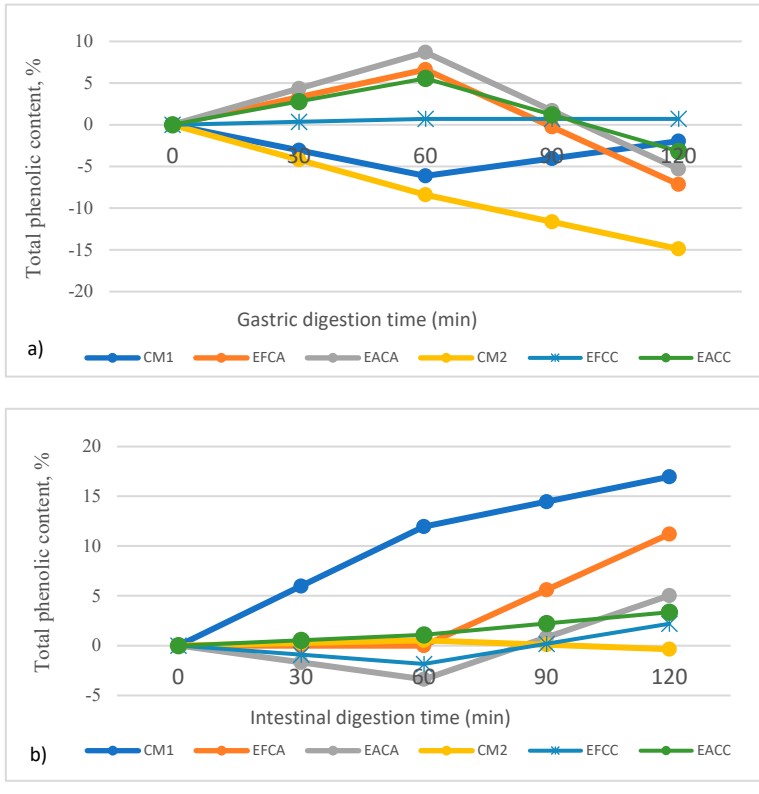

**Figure 2.** In vitro gastric (**a**) and intestinal digestion (**b**) of remanent phenolic content in sweet potatoes samples combined with herbal aqueous extract.

$CM_1/CM_2$ are sweet potatoes purees treated by steam and hot air convection; EFCA and EACA are steamed sweet potatoes samples mixed with fennel or anise aqueous extract; EFCC and EACC are baked sweet potatoes samples combined with fennel or anise aqueous extract.

According to Figure 2a, the variation of total phenolic content of all the samples, except for EFCC, started to generate a negative downward slope after 1 h of simulated gastric digestion. Regarding EFCC, there was a release of phenols from the matrix throughout the gastric phase, reaching a final concentration of $0.7 \pm 0.07\%$ after two hours of digestion.

Figure 2b illustrates that at the end of the in vitro digestion, five out of six samples registered positive values ranging from $2.2 \pm 0.53$ to $16.95 \pm 1.08\%$. In our previous study [9] regarding zucchini purees enriched with fennel or anise aqueous extract, the remanent phenolic content registered after 4 h simulation of digestion showed values ranging between $0.63 \pm 1.57$ and $9.20 \pm 1.19\%$. This fact might be due to the different nutritional composition of each vegetable studied.

Similar to Liu et al. [30], the polyphenols dynamics during the gastrointestinal digestion are correlated with the small or large particle disruption. These are directly induced by the mechanical actions and the activity of gastric fluids, and this could affect the speed of the bioactive compounds release. The thermal treatment and the aqueous extract type could also influence the digestion of the sweet potato purees. The digestion dynamic could be influenced by the stage and the phenolic structure's possibility to be changed by glycosylation, hydroxylation, and dimerization, as stated by Ziółkiewicz et al. [31].

### 3.4. Influence of Heat Treatments on Color Parameters of Sweet Potatoes Purees

The color of fruit and vegetable foods represents a key element in consumers' choice and the success of these products [26,32]. The color parameters of orange sweet potato puree samples enriched with anise or fennel aqueous extract for the evaluated heat treatments are summarized in Table 3. In our research, after cooking (baking and steaming), L* decreased (from $54.59 \pm 0.00$ (raw puree sample) to $47.51 \pm 0.06$ (EFCC sample)), a* decreased (from $25.37 \pm 0.00$ (control sample) to $19.84 \pm 0.15$ (EACC sample)), and b* increased (from $40.2 \pm 0.00$ (raw puree sample) to $42.22 \pm 0.13$ (EACC sample)). Zhang et al. [26] also reported similar color changes in the case of sweet potato puree processed over 100 °C. According to Gallego-Castillo et al. [33], the decrease in lightness may be correlated with mechanical destruction produced by the blending process used to obtain sweet potato puree, because this can deteriorate the weak chlorophyll and carotenoid membranes, thus determining their oxidative or enzymatic degradation. The increase of blue–yellow components after cooking may be due to the development of brown color compounds by the Maillard reaction or the reduction of the red–green components during cooking. After the heat treatments, the total color difference (ΔE) values increased for all samples, a fact that could be attributed to the increase of bioactive compounds from the fennel or anise aqueous extract.

The Chroma (C*), a measure of the intensity of color, ranged from $47.54 \pm 0.00$ (control sample) to $45.78 \pm 0.05$ (EACA sample). Slight changes were observed, after cooking, compared with the control sample. The hue angle (h*) values increased during cooking and ranged from $32.26 \pm 0.00$ (control sample) to $64.82 \pm 0.09$ (EACC sample). This increase was linked to the blue–yellow component (b*) values. According to [34], these values suggest the stable color of sweet potato puree enriched with herbal aqueous extract, in orange–red angle (when Hue < 90 °C). Table 3 shows the color changes of sweet potato samples combined with herbal aqueous extract during storage. The lightness (L*) of sweet potato puree enriched with anise or fennel aqueous extract decreased during the storage time. After 7 days of storage, the L* values of samples with anise or fennel aqueous extract were lower compared with the raw sample. The decrease of b* values after 7 days of storage could be associated to some color compounds loss because these pigments are sensitive to heat treatment, light, and oxygen. The findings for thermally treated orange sweet potato samples in the present research were similar to Grace et al. [35].

**Table 3.** Influence of thermal treatments on color parameters of sweet potatoes samples combined with herbal aqueous extract.

| | Raw | EFCA | EACA | EFCC | EACC |
|---|---|---|---|---|---|
| | | | **$T_0$** | | |
| L* | 54.59 ± 0.00 [A] | 49.59 ± 0.02 [B] | 49.07 ± 0.00 [C] | 47.51 ± 0.06 [E] | 47.81 ± 0.11 [D] |
| a* | 25.37 ± 0.00 [A] | 21.13 ± 0.01 [C] | 20.15 ± 0.00 [D] | 21.32 ± 0.03 [B] | 19.84 ± 0.15 [E] |
| b* | 40.2 ± 0.00 [D] | 41.59 ± 0.20 [B] | 41.11 ± 0.06 [C] | 41.76 ± 0.02 [B] | 42.22 ± 0.13 [A] |
| ΔE | - | 6.70 ± 0.05 [D] | 7.65 ± 0.01 [C] | 8.30 ± 0.06 [B] | 8.97 ± 0.14 [A] |
| C* | 47.54 ± 0.00 [A] | 46.65 ± 0.18 [B] | 45.78 ± 0.05 [C] | 46.88 ± 0.03 [B] | 46.65 ± 0.18 [B] |
| h* | 32.26 ± 0.00 [D] | 63.06 ± 0.10 [C] | 63.89 ± 0.03 [B] | 62.95 ± 0.02 [C] | 64.82 ± 0.09 [A] |
| WI | 34.26 ± 0.00 [A] | 31.31 ± 0.13 [C] | 31.51 ± 0.04 [B] | 29.61 ± 0.02 [E] | 30.0 ± 0.04 [D] |
| BI | 152.41 ± 0.00 [C] | 178.78 ± 1.33 [B] | 177.46 ± 0.38 [B] | 192.94 ± 0.22 [A] | 192.15 ± 0.39 [A] |
| YI | 105.2 ± 0.00 [C] | 119.83 ± 0.61 [B] | 119.68 ± 0.17 [B] | 125.57 ± 0.10 [A] | 126.13 ± 0.11 [A] |
| | | | **$T_7$** | | |
| L* | 54.59 ± 0.00 [A] | 44.64 ± 0.20 [B, C] | 44.95 ± 0.51 [B] | 44.79 ± 0.43 [B, C] | 44.03 ± 0.30 [C] |
| a* | 25.37 ± 0.00 [A] | 16.74 ± 0.23 [C] | 17.0 ± 0.02 [C] | 17.94 ± 0.16 [B] | 17.00 ± 0.03 [C] |
| b* | 40.2 ± 0.00 [A] | 33 ± 0.06 [C] | 34.19 ± 0.84 [C] | 37.93 ± 0.70 [B] | 36.66 ± 0.18 [B] |
| ΔE | - | 15.01 ± 0.24 [A] | 14.12 ± 0.71 [A] | 12.52 ± 0.55 [B] | 13.94 ± 0.25 [A] |
| C* | 47.54 ± 0.00 [A] | 37.0 ± 0.05 [D] | 38.18 ± 0.76 [D] | 41.95 ± 0.69 [B] | 40.41 ± 0.15 [C] |
| h* | 32.26 ± 0.00 [C] | 63.1 ± 0.36 [B] | 63.55 ± 0.53 [B] | 64.69 ± 0.21 [A] | 65.13 ± 0.14 [A] |
| WI | 34.26 ± 0.00 [A] | 33.41 ± 0.14 [B] | 33.0 ± 0.02 [C] | 30.65 ± 0.08 [E] | 30.96 ± 0.15 [D] |
| BI | 152.41 ± 0.00 [C] | 147.17 ± 0.98 [D] | 153.07 ± 2.46 [C] | 179.58 ± 2.28 [A] | 174.30 ± 0.66 [B] |
| YI | 105.2 ± 0.00 [C] | 105.61 ± 0.67 [C] | 108.65 ± 1.43 [B] | 120.96 ± 1.06 [A] | 118.96 ± 0.21 [A] |

The averages on the same line with different superscripts (A, B, C, D and E) are statistically significantly different ($p < 0.05$). $T_0$—samples on the first day of storage; $T_7$—samples after 7 days refrigeration storage.

### 3.5. Rheological Analysis

The results of the rheological analysis show lower values for loss modulus than for storage modulus for the same puree sample. This relationship between the moduli indicates a predominant elastic character and is specific to vegetal puree [36]. In Figure 3, it can be seen that the type of thermic treatment influenced the values of the rheological parameters. The hot air-treated samples, as well as the ones with added aqueous extracts, presented higher values of G′ and G″ when compared to those obtained by steam convection. This may be due to the higher amount of disponible water which softens the vegetal tissue [33]. In the hot air-treated samples, the complex matrix of the puree samples is determined mainly by the interactions between gelatinized starch, pectin, and water, causing modifications. Consequently, after 7 days of storage, the differences between the rheological moduli become more evident and the samples may be clearly individualized by thermic treatment. This behavior could be observed both in strain and frequency tests. Similar rheological values have been reported for different types of sweet potato puree by several authors [17,37,38].

The scientific literature states that due to their structural complexity, vegetal purees can present both pseudoplastic and thixotropic behavior, depending on time and certain conditions [18]. In this context, the thixotropy of the fresh and stored puree samples was investigated by a three interval thixotropy test. The recovery degrees presented in Figure 4 show a good capacity of recovering the stress induced modifications, since they registered values between 81.3 and 89.5%.

The averages for each group of columns with different superscripts are statistically significantly different ($p < 0.05$).

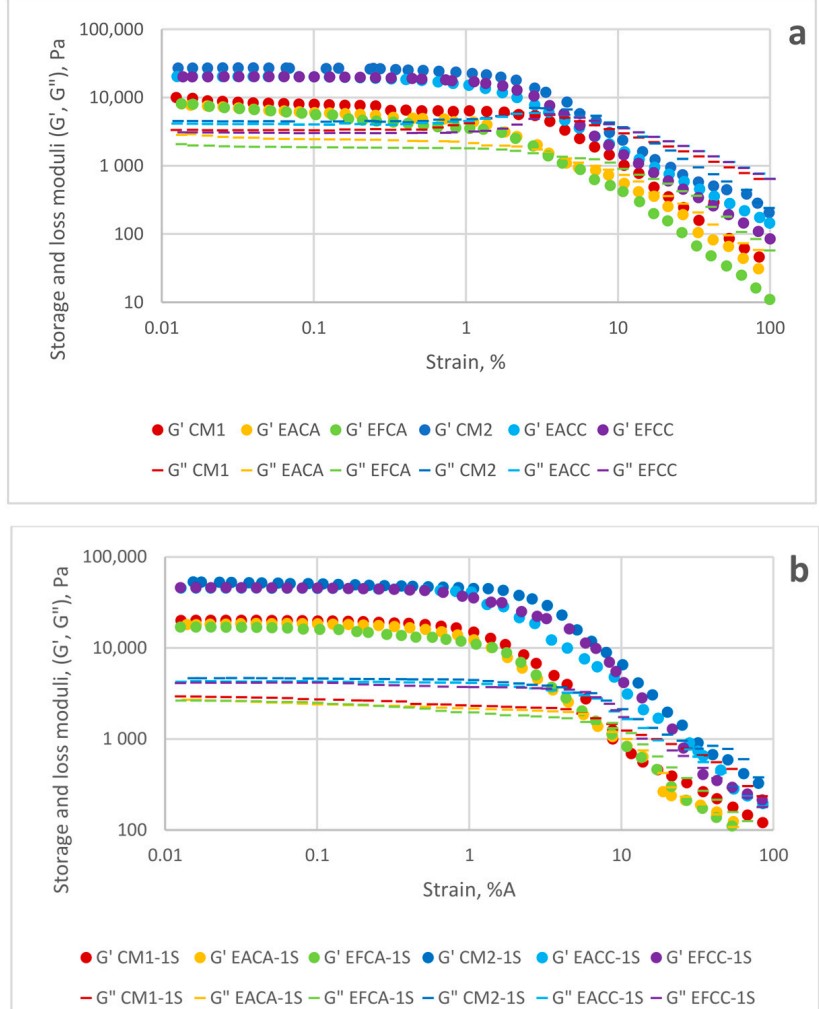

**Figure 3.** *Cont.*

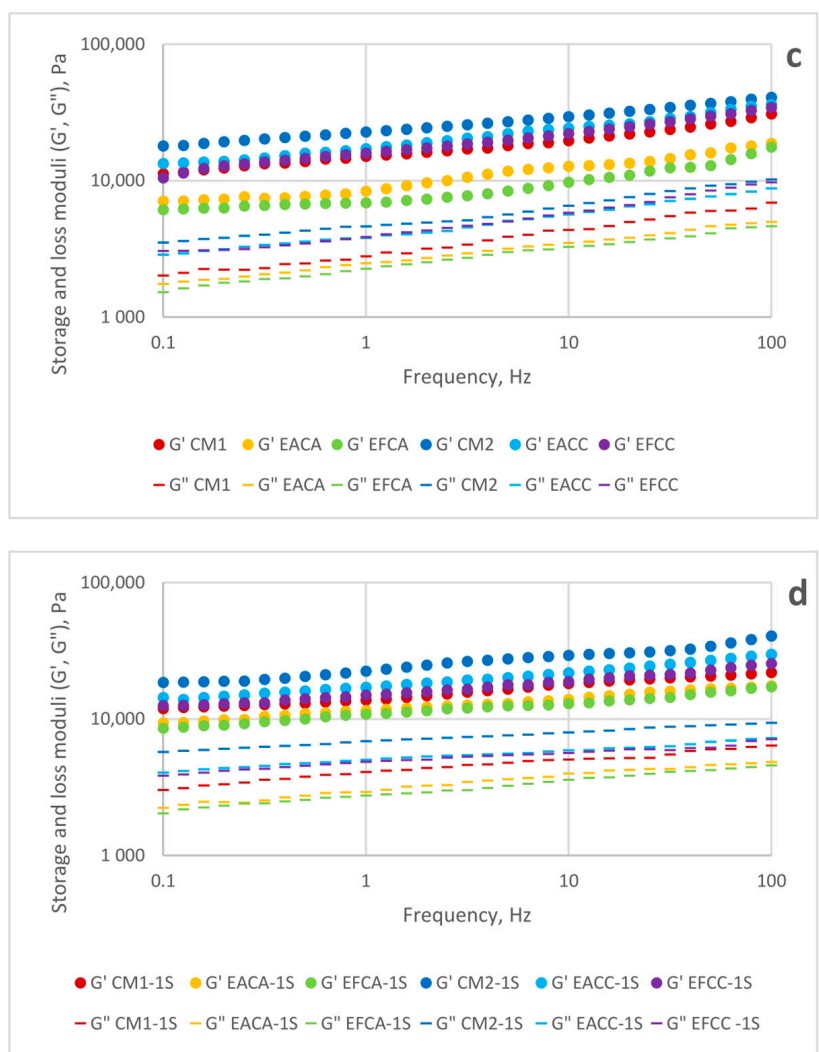

**Figure 3.** Rheological behaviour of fresh (**a**,**c**) and stored (**b**,**d**) samples measured by strain sweep (**a**,**b**) and frequency sweep (**c**,**d**) tests.

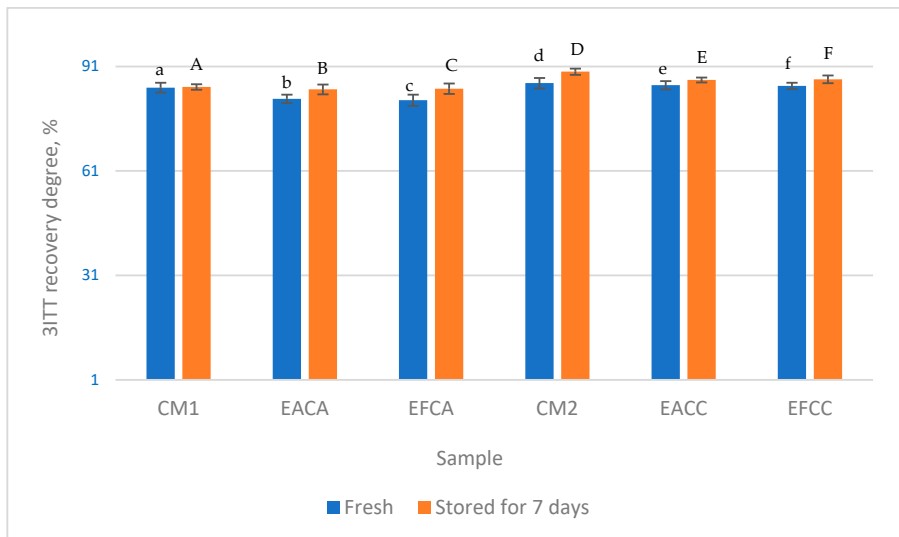

**Figure 4.** The 3ITT recovery degrees of sweet potatoes purees, supplemented with 6% aqueous anise/fennel extract, analyzed on the first and 7th day of storage. The averages for each group of columns with different superscripts are statistically significantly different ($p < 0.05$).



### 3.6. Instrumental Determination of Texture

Table 4 illustrates the results of the instrumental texture analysis. It could be remarked that the firmness is influenced by the thermal treatment applied and by the supplementation with extract. Hot air convection determined an increase in firmness with nearly 3N, both in fresh and stored samples. Moreover, this type of treatment determines increases in all textural parameters: nearly 50% for adhesiveness, 65% for cohesiveness, and 60% for springiness. Steam convection makes an important amount of water available which, when combined with heat above 66 °C [19], leads to hydration of starch granules and the increase of their size and weight.

The texture parameters did not significantly vary during 7 days of storage, indicating the stability of the sweet potato purees samples in this interval.

**Table 4.** Influence of hot air or steam convection over textural parameters of sweet potatoes purees samples.

| Textural Parameters | $T_0$ | | | | | |
|---|---|---|---|---|---|---|
| | CM$_1$ | EFCA | EACA | CM$_2$ | EFCC | EACC |
| Firmness, N | 5.71 ± 0.11 [B,C] | 3.59 ± 0.10 [C] | 4.27 ± 0.69 [C] | 13.24 ± 1.21 [A] | 9.87 ± 0.95 [A] | 9.52 ± 3.26 [A,B] |
| Adhesiveness, mJ | 23.16 ± 3.12 [B] | 35.68 ± 3.57 [A,B] | 38.86 ± 3.79 [B] | 50.24 ± 4.16 [A,B] | 91.25 ± 3.99 [A] | 98.87 ± 4.50 [A] |
| Cohesiveness | 0.53 ± 0.04 [A] | 0.69 ± 0.02 [A] | 0.60 ± 0.06 [A] | 0.51 ± 0.01 [A] | 0.66 ± 0.09 [A] | 0.65 ± 0.03 [A] |
| Springiness, mm | 8.69 ± 0.03 [B] | 13.13 ± 0.15 [A] | 13.31 ± 0.66 [A] | 8.72 ± 0.52 [B] | 13.92 ± 1.25 [A] | 14.06 ± 0.91 [A] |
| | $T_7$ | | | | | |
| Firmness, N | 5.70 ± 0.25 [B] | 3.30 ± 0.25 [C] | 3.96 ± 0.13 [C] | 13.63 ± 0.88 [A] | 9.44 ± 0.77 [B] | 8.98 ± 0.21 [B] |
| Adhesiveness, mJ | 23.61 ± 2.25 [A,B,C] | 33.77 ± 5.15 [B,C] | 20.44 ± 8.90 [C] | 45.9 ± 15.31 [A] | 88.01 ± 5.11 [A,B] | 91.31 ± 4.46 [A,B,C] |
| Cohesiveness | 0.39 ± 0.02 [A] | 0.58 ± 0.13 [A] | 0.54 ± 0.05 [A] | 0.44 ± 0.07 [A] | 0.47 ± 0.14 [A] | 0.60 ± 0.02 [A] |
| Springiness, mm | 8.05 ± 0.22 [A] | 12.65 ± 1.30 [A] | 13.76 ± 0.85 [A] | 8.71 ± 2.74 [A] | 12.23 ± 1.26 [A] | 13.10 ± 0.30 [A] |

The averages on the same line with different superscripts (A, B, and C) are statistically significantly different ($p < 0.05$).

### 3.7. FT-IR Spectroscopy

From the FT-IR spectra evaluation presented in Figure 5, similar peaks and bands to those determined in a similar study on zucchini of [9] could be observed. Notably, 3315.85–3323.06 cm$^{-1}$ band corresponding to water presence, 1418.18–1458.88 cm$^{-1}$ specific to trans-anethole, 1635–1638 cm$^{-1}$ related to estragole, and 1027–1030.37 cm$^{-1}$ which corresponds to cellulose, hemicellulose, and pectin are visible in the FT-IR spectra for all the sweet potato puree samples. Similar findings were presented in a study regarding 10 varieties of sweet potato [39].

It seems that both hot air and steam convection are not changing the main compounds from the matrix, as well as the added aqueous extracts.

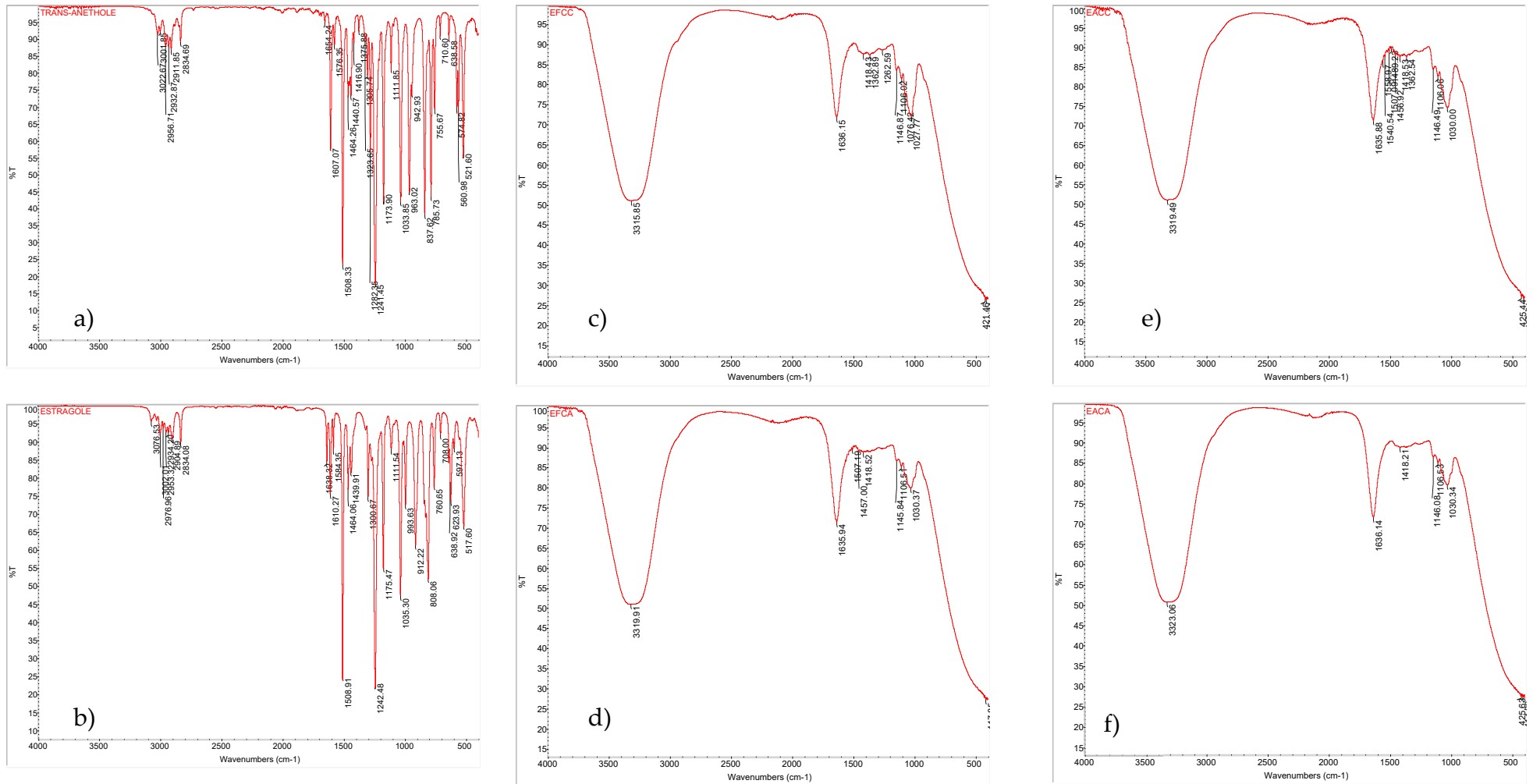

**Figure 5.** FT-IR spectroscopy spectra of (**a**) standard trans-anethole; (**b**) standard estragole [40]; (**c**) EFCC—baked sweet potatoes puree mixed with fennel aqueous extract; (**d**) EFCA—steamed sweet potatoes puree mixed with fennel aqueous extract; (**e**) EACC—baked sweet potatoes puree mixed with anise aqueous extract; (**f**) EACA—steamed sweet potatoes puree mixed with anise aqueous extract.

*3.8. Sensory Analysis*

The results of the sensory evaluation of orange sweet potato puree enriched with galactogogue aqueous extract and processed with different cooking methods are presented in Figure 6a. The color attribute ranged from 7.5 to 8.3 with control samples as least preferred and steamed sample EACA as most favored. For attributes of taste and aroma, greater acceptability was remarked for baked sweet potatoes puree mixed with anise aqueous extract (encoded EACC), compared to other samples.

The aftertaste and mouthcoating sensation were assessed with the highest scores of 8.3 and 8.2 for EACC sample and were judged as a long-lasting and persistent orosensation. Similar results were reported for asparagus by [41]. The sensory attributes which define the texture of the orange sweet potato puree enriched with herbal aqueous extract, characterized by firmness, consistency, and cohesiveness, ranged from 6.3 (EFCC sample) to 7.1 (EACC sample). The EFCC and EACC samples were processed by hot air convection and were the most preferred. The highest overall acceptability of orange sweet potato puree enriched with herbal aqueous extract ranged from 7.5 to 8.5 and was attributed to the hot air convection.

The sensory analysis data for all the orange sweet potato puree samples did not show a significant difference between sensorial attributes. According to our results, the panelists preferred the sweet potatoes puree with anise aqueous extract, processed at a higher temperature (180 °C) and for a long time (35 min).

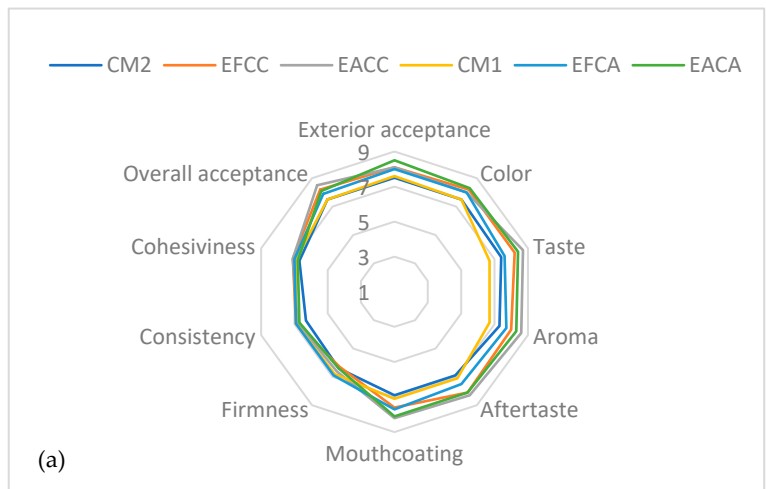

(a)

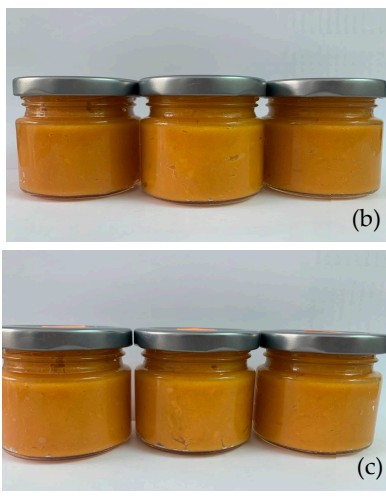

(b)

(c)

**Figure 6.** (**a**) Comparative chart of the sensory evaluation; (**b**) baked sweet potato purees with aqueous herbal extract; (**c**) steamed sweet potato aqueous herbal extract.

## 4. Conclusions

Sweet potatoes are a crop becoming recognized as a healthy food according to their variety of metabolites, especially antioxidants. This study showed that thermal treatments had a different influence on phytochemical compounds and antioxidant activity. Processing time could be reduced, and the retention of phytochemicals could be considerably increased by using water vapor convection. As well, steaming proved to be a milder treatment regarding colour parameters. As extensively mentioned in the scientific literature, brighter (higher L*) products increase the customers' acceptability. In vitro digestibility marked the existence of a remanent phenolic content after a 4 h simulation. In the case of the textural parameters, no significant differences were registered between fresh and stored samples, proving that the seven days of storage are proper for sweet potato purees combined with different aqueous extracts. Notably, the FT-IR spectra have identified the presence of galactogogue compounds, represented by trans-anethole and estragole. Different potential use of DSC findings could apply to further types of sweet potato processing.

The present results can facilitate a better comprehensive understanding of the impact that baking and steaming processing has on bioactive compounds in sweet potatoes and maybe contribute to the improvement of ready-to-eat product processing technology. The consumption of sweet potatoes, but also of medicinal herbs, is known to bring numerous health benefits (antioxidant, antimicrobial, and other properties). Therefore, adding such ready-to-eat products to the diet could bring a positive impact to consumers. Further clinical tests should be considered regarding the efficiency of the consumption of lactation-stimulating products.

**Author Contributions:** Conceptualization, E.B. and O.-V.N.; methodology, L.-A.T.; software, G.-D.M.; validation, D.-G.A., O.-V.N. and L.-A.T.; resources, O.-V.N.; data curation, G.-D.M.; writing—original draft preparation, L.-A.T.; writing—review and editing, O.-V.N.; visualization, D.-G.A.; supervision, E.B. and B.I.Ș. All authors have read and agreed to the published version of the manuscript.

**Funding:** The authors received for technical and financial support from project GI 9186/29.03.2023.

**Data Availability Statement:** Not applicable.

**Acknowledgments:** The Integrated Center for Research, Expertise and Technology Transfer in the Food Industry, as well as the MoRAS Center developed through the POSCCE ID 1815, SMIS number 48745 (www.moras.ugal.ro, accessed on 16 April 2022), are to be thanked for their technical assistance throughout this experiment.

**Conflicts of Interest:** The authors declare no competing interest.

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
