# Peer review of "Sweet Potatoes Puree Mixed with Herbal Aqueous Extracts: A Novel Ready-to-Eat Product for Lactating Mothers"

_processes, doi:10.3390/pr11072219_

Round 1

Reviewer 1 Report

The paper is well-written and interesting.

I recommend its publication in the current form, though I suggest authors to use in the future a larger group for testing as for statistical evaluations 10 people is not a sample big enough.

I also suggest to use mothers instead of moms in the Abstract.

The paper needs only small correction, English is fine.

Author Response

The authors would like to thank you for the suggestions and the time spent reviewing the manuscript.

Reviewer 2 Report

The authors presented work on the development of a food product that could impact breastfeeding moms. However, the title seems a bit misleading and irrelevant as it includes 'destination'. I reviewed the paper and made the major comments below. The M&M  section needs to be revised as well.    Title: I do not agree with the last words- special destination. You have not developed this product for a specific region. Suggest the title could be- Sweet Potatoes Puree Mixed with Herbal Aqueous Extracts: a Novel Ready-To-Eat Product   Abstract: The first two sentences are not relevant. They could go with an article on infant formula. I suggest removing them from there. L(ine) 16- Add 'fortified' or 'mixed' before .....with.. L18- Replace 'used' with 'examined' or 'investigated'. L20- Replace 'Between' with "among". L45- Correct the ref style. Follow the guideline of the journal. It could be- According to Sibeko and Johns [5]...................... OR, A significant ........................[5]. This is a major correction that needs to be made as it appears in the multiple places of this manuscript.  L51- Correct the ref style. L61- Correct the ref style. No need to use the page number.  L69- Add 'mixed' before 'with'. L73-78- This paragraph might not be necessary and seems irrelevant. This also has a confusing word 'destination'. What does the destination refer to? Are you trying to develop a product for a specific destination, e.g; Europe, Asia, or any specific country or region or group of people? Plus, you already mentioned the scientific aspects in L71-72. Therefore, I suggest removing this paragraph. Section 2.1- L81-82- List all (or most of) the chemicals and the supplier/ source in addition to the reference. The way you expressed/ explained is not acceptable. Section 2.2.1- L84-85- Brief description of the prep is required. Plus the ref. L88- Revise this statement. It could be- Sweet potatoes (4-5 cm diameter) were collected..... L93-94- Keep a space before the degree sign. L97- What does the 1900 rot/min mean? Is it 1900 rpm? Section 2.2.3- L101-103- Use the full name of CM, EFCA, EACA, EFCC and EACC. This is where you should clarify and detail your coding so that the readers get them easily. Section 2.3.1- L106-108- use proper ref style. Write the equation and cite the ref. Section 2.3.2- No. Please read other papers and look at how the authors describe the M&M section. Brief description and cite the ref (a general rule!). Section 2.4-L115-116- Revise the statement.  L121- Brief description plus cite the ref related to the carotenoid measurement. Section 2.5- Follow the previous comment. { Overall, revise/ rewrite the m&M section} L175- Delete L180- Correct the ref style. L193- Correct the ref style. L220- Fig 1- Y axis could start from 0 or 10, there is nothing to show the +ve Y value. Right? Similarly, X value could start from -20 or similar? L226- Correct the ref style. L277-279- Are these necessary? You already described them earlier. Section 3.3- L271-289- Explain how-why does the phenolic content show up/down trend. What is the science behind it? L351-352- You already explained these codes. L361-Fig 4-Improve the bar graph. Y values could be with intervals of 30 (1, 31, 61, 91) L363-364- You already have explained the codes. Overall, how could the product help lactation in breastfeeding moms? What could be the science and mechanistic approach of the product in achieving such targets? Very few data have been compared with the literature.    Conclusion: Please include some possible impacts of the product that could happen on human health and nutrition too along with the direction of future research.   

A number of sentences could be written in active voice. In several places, sentence structures were poorly formed. Suggest taking support from English editing. 

Author Response

(The authors gave the same response as above.)

Reviewer 3 Report

·         The whole paper should be checked for language (with some language tool like Instatext, Gramarly, etc.)

·         Lines 45/51/69, etc.(whole paper) - According to (5)- the name of the author/s should be written (e.g. Brown (5) or Brown et al. (5), etc.

·         Vegetal material (line 73).- prefere „plant-based“

·         Line 74-you didnt treat crop- tuber or potatoe

·         2.2. Sample preparation should be described. Same for 2.3.1., 2.3.2., etc.

·         2.4.- from the paper it is not evident which antioxidant assay was used, same from the table 2 (we see units but not the method- ABTS, DPPH, ORAC, etc?). Also, the whole this section should be described better (e.g. for carotenoids it should be stated which methodology and techniques were used – spectrophotometry/HPLC, etc. without openening and reading other papers).

·         Line 99: only fennel or anis extract also?

·         2.2.3. should be incorporated in 2.2.2

·         „n“ in n-hexane should be written in italic

·         Methods in 2.4-2.9 should be written in more details with adequate references.

·         Line 175- this?

·         Table 2 and 3- T0 and T1 should be fully written in table (no abbrevation and definition in caption)

·         Results for other phytochemicals and antioxidant activity apart from carotenoids should be discusssed better

This study is well planned and lot of work has been performed. However, there is a great need for the improvement of some written sections, especially Materials and Methods. Also, the whole discussion could be written better with adding more references and results/conclusions from the similar studies.

The whole paper should be checked for language (with some language tool like Instatext, Gramarly, etc. or by native English speaker)

Author Response

(The authors gave the same response as above.)

Round 2

Reviewer 2 Report

Thanks for addressing the comments. Some additional comments are-

1) Since you developed the product for a certain group of people (lactating moms), it would be nice to add them in the title. Title could be- Sweet Potatoes Puree Mixed with Herbal Aqueous Extracts: a Novel Ready-To-Eat Product for Lactating Mothers (or Women)

2) Check the reference style again. I suggest the ref style should not contain the year of publication as you already numbered them. e.g; Sibeko et al. [5] 

3) The English grammar of the manuscript must be improved.

The English grammar of the manuscript must be improved. Suggest taking editing service. 

Author Response

(The authors gave the same response as above.)

Reviewer 3 Report

All suggestions have been more or less accepted and the manuscript has been improved.

Author Response

Thank you for your remark!